# Light-Transmitting Lithium Aluminosilicate Glass-Ceramics with Excellent Mechanical Properties Based on Cluster Model Design

**DOI:** 10.3390/nano13030530

**Published:** 2023-01-28

**Authors:** Minghan Li, Chuang Dong, Yanping Ma, Hong Jiang

**Affiliations:** 1State Key Laboratory of Marine Resources Utilization in South China Sea, Hainan University, Haikou 570228, China; 2Special Glass Key Lab of Hainan Province, Hainan University, Haikou 570228, China; 3School of Materials Science and Technology, Dalian Jiaotong University, Dalian 116028, China

**Keywords:** lithium aluminosilicate glass-ceramics, cluster-plus-glue-atom model, average cation valence, light-transmitting glass-ceramics, mechanical property, structural analysis

## Abstract

In this study, for the first time, a cluster-plus-glue-atom model was used to optimize the composition of lithium aluminosilicate glass-ceramics. Basic glass in glass-ceramics was considered to be a 16-unit combination of three-valence {M_2_O_3_} and one-valence {Li_2_O} units. By adjusting the ratio of {M_2_O_3_} and {Li_2_O}, the composition of basic glass could be optimized. After optimization, the average cation valence of the base glass was increased to 2.875. After heat treatment of the optimized base glass, it is found that the crystal size, proportion, and crystallinity changed obviously compared with that before optimization. The main crystalline phases of all the lithium aluminosilicate glass-ceramics prepared in this work were Li_2_Si_2_O_5_ and LiAlSi_4_O_10._ All optimized glass-ceramics had an obvious improvement in the crystallinity, with one of the largest having a crystallinity of over 90%. Furthermore, its bending strength was 159 MPa, the microhardness was 967 Hv, and the visible light transmission rate exceeded 90%. Compared with the widely used touch panel cover glass, the optical properties were close, and the mechanical properties were greatly improved. Due to its excellent performance, it could be used in microelectronics, aerospace, deep-sea exploration, and other fields.

## 1. Introduction

Glass-ceramics are polycrystalline solids obtained due to controlled crystallization of the glass. They are obtained by uniformly precipitating numerous tiny crystals in a glass matrix to form a multiphase composite comprising dense crystalline phases and glass phases. They exhibit remarkable hardness and favorable mechanical, thermal, and chemical stabilities [1,2]. The properties of glass-ceramics are determined by those of the glass matrix and precipitated crystals, for example, glass-ceramics with a *β*-quartz solid solution as the main crystalline phase have a low thermal expansion coefficient and high transparency [3]. The low thermal expansion coefficient of lithium aluminosilicate glass-ceramics is due to the special structure of the precipitated phases; if the grain size is smaller than the wavelength of the visible light, the glass-ceramics are transparent.

Among the various existing glass-ceramics, the lithium aluminosilicate system (Li_2_O–Al_2_O_3_–SiO_2_) has been extensively studied owing to its excellent mechanical properties, low coefficient of thermal expansion, and high thermal impact resistance [4,5]. By controlling the type, number, and size of the precipitated crystals, highly transparent glass-ceramics, which have become the material of choice for manufacturing cookware, high-temperature observation windows, laboratory heating appliances, and chemical pipes, among others, can be prepared [6]. Kumar et al. [7] prepared lithium aluminosilicate glass-ceramics with a *β*-quartz solid solution as the main crystalline phase, exhibiting a very low coefficient of expansion and a visible light transmission rate of approximately 90%. Corning and Schott companies also had lithium aluminosilicate glass-ceramics products with a similar structure and performance, and patents were pending on them [8,9,10,11]. Hui Zhang et al. [12] prepared a lithium aluminosilicate glass-ceramic with excellent mechanical properties. Li_2_Si_2_O_5_ rods were successfully synthesized by a modified two-step solid-state reaction sintering method. The Li_2_Si_2_O_5_ rods were used to strengthen and toughen the homologous Li_2_Si_2_O_5_ glass-ceramic, which indicates a high flexural strength and toughness of 315 ± 24 MPa and 2.42 ± 0.6 MPa·m^1/2^, respectively. The main crystalline phase of the lithium aluminosilicate glass-ceramics prepared by Dittmer et al. was Li_2_Si_2_O_5_, which had an interlocking structure [13]. The interlocking structure improved the mechanical properties of the glass-ceramics. Currently, it is possible to prepare different crystalline forms of lithium aluminosilicate glass-ceramics, and differences in the precipitated crystal type, crystallinity, and size affect the properties of these glass-ceramics.

When designing and optimizing the composition of glass-ceramics, the type of crystals to be precipitated is usually determined first, and then the composition of the base glass is developed using the phase diagram. Konar and Aliyah [14,15] used a ternary phase diagram to design the composition of lithium aluminosilicate base glasses, which could be used to roughly determine the component ratios of Li_2_O, Al_2_O_3_, and SiO_2_ in the glasses. The proportion of specific components was then adjusted through several experiments. Building on these previous results, numerous researchers have improved the base formulation of glass-ceramics, and the influence on the crystallization of the composition has been extensively investigated. For example, Ananthanarayanan and Glatz [16,17] used magic angle spinning nuclear magnetic resonance (MAS-NMR), X-ray diffraction (XRD), and other techniques to study lithium aluminosilicate glass-ceramics, and their results complemented each other, providing an in-depth analysis of the influence of composition on crystallization. However, regardless of whether glass-ceramics are obtained based on previous experience or by designing the base composition using phase diagrams, extensive experiments are required to determine the glass formulation. Therefore, there is an urgent need to introduce theoretical models to guide composition design. In this paper, we choose the cluster-plus-glue-atom model to optimize the base glass formulation to obtain excellent performance in glass-ceramics.

## 2. Cluster-Plus-Glue-Atom Model

The main theories describing glass as an amorphous compound are the crystallite theory and random network theory. However, neither of these theories can accurately describe the structure of glass because of its random and disordered structure. Thus, these theories can only be used as qualitative guides to design glass compositions. It is possible to model the relationship between the structure and properties for single-crystal and polycrystalline materials. However, until recently, the relationship between composition, structure, and properties of glasses as amorphous materials remained unknown [18]. This is mainly because current studies lack a structural carrier to accurately define the composition of glasses, making it impossible to establish a correlation between the glass composition, structure, and properties, and to develop a corresponding quantitative theory for composition design. It is well known that material properties depend on the material composition, which in turn depends on the material structure. Therefore, it is crucial to study the structural carriers that dictate the glass composition from their structure to provide a practical and quantitative composition design method from an atomic perspective.

In this study, we introduce, for the first time, a cluster-plus-glue-atom model for determining the optimal composition of glass-ceramics. This model has been extensively applied in quasicrystals [19,20], metallic glasses [21,22], solid solution alloys [23], and inorganic nonmetals [24], effectively guiding the composition design of related materials. Furthermore, the relationship between composition design and crystallization has been discussed in the studies related to metallic glass [25,26,27]. Based on the model, a structural unit comprises a coordination polyhedron covering the nearest neighbors (known as a cluster) and a few second nearest-neighbor glue atoms, formulated as [cluster](glue atom)_x_. To design a glass composition, this model defines the structural carrier of the composition-molecule-like structural units. A molecule-like structural unit is the smallest structural unit of a substance and carries both its structural characteristics and compositional information. Conventional chemical formulas do not represent the smallest individual structural units. For example, in beta-tridymite (Figure 1, any polymorphs of silica gives the same structure unit), the structure unit is [Si–O_4_]Si = {Si_2_O_4_}, that is, the [Si–O_4_] tetrahedral cluster plus one Si as the glue atom [28]. The entire *beta-tridymite* is then equivalent to a stack of {Si_2_O_4_} structural units, such that their structure is divided into two parts: one part is the molecular structure unit {Si_2_O_4_}, and the other part is the stacking mode of the unit in space. From the analysis of the crystal structure, it was clear that {Si_2_O_4_} was stacked in a dense hexagonal arrangement. Considering each {Si_2_O_4_} base structure unit as a hard sphere, this dense hexagonal stacking arrangement results in the formation of a supercluster structure unit containing 16 hard spheres, each representing one basic structure unit {Si_2_O_4_}) [29]. The supercluster structure unit of *β*-SiO_2_ was formulated as [{Si_2_O_4_}−{Si_2_O_4_}_12_ ] {Si_2_O_4_}_3_ = {Si_2_O_4_}_16_.

By analyzing the composition of commercially available glasses, we found that the average cation valency of common silicate white glasses was between three and four. Therefore it is anticipated that various glasses are mixtures of a three-valence {M_2_O_3_} units and four-valence {Si_2_O_4_} units, where M represents an average three-valence cation, calculated by averaging the valences of cations of difference valences (for example, one two-valence Mg is matched with one Si to obtained average cations of three valence, M^3+^_2_ = Mg^2+^_1_Si^4+^_1_).

To facilitate understanding, we first clarify the algorithm for the average cation valency. The average cation valence is derived from the valence of various cations in the base glass and their corresponding molar ratios. For example, in the simplest ternary lithium aluminosilicate glass system, where Li_2_O, Al_2_O_3_, and SiO_2_ are added in ratios of x, y, and z mol%, respectively, the average cation valence is:$$\mathrm{average}\;\mathrm{cation}\;\mathrm{valence}=\frac{2x+6y+4z}{x+y+z}$$

For most high-aluminosilicate glasses, including the Corning and Schott series, the average valence of cations is already close to the lower limit valence of three. Thus, these glasses can be described as having an average cation valence of three for {M_2_O_3_}_16_. As glass-ceramics should have a slightly weak glass-forming ability, in this study, it was suggested that the average cation valency of the base glasses should be reduced to less than three. Particularly, for the lithium aluminosilicate system, glass-ceramics are considered to have a 16-unit combination of three-valence {M_2_O_3_} and one-valence {Li_2_O} units, thus completing the quantitative composition design of glass-ceramics. For other systems of glass and glass-ceramics, the model is used in the same way. Such as CaO–2SiO_2_, SrO–2SiO_2_, BaO–2SiO_2_ [30,31,32,33], CaO–MgO–SiO_2_ [34], and so on. Taking CaO–MgO–SiO_2_ glass-ceramics as an example, it can be considered to be a 16-unit combination of three-valence {M_2_O_3_} and two-valence {Mg_2_O_2_} units for design and optimization.

## 3. Experiments

### 3.1. Formulations and Composition Design

A 16-unit mixture of the {M_2_O_3_} structural units, with M being an average 3-valence cation, and {Li_2_O} structural units with a valency of 1 were used as the basis for the composition design of the base glasses. The base glass C1 formulation was obtained from our previous experimental results. The specific idea is to set the main crystalline phases of glass-ceramics as Li_2_Si_2_O_5_ and LiAlSi_4_O_10_, and to determine the range of SiO_2_, Al_2_O_3_, and Li_2_O additions using a Li_2_O–Al_2_O_3_–SiO_2_ ternary phase diagram. P_2_O_5_ and ZrO_2_ were selected as nucleating agents. After a series of experiments, the ratio of these components was adjusted to obtain the C1 formulation. However, after crystallization, the C1 glass had coarse grains and low crystallinity. Therefore, the C1 formulation was analyzed using the cluster-plus-glue-atom model to determine the composition of M in {M_2_O_3_}. Subsequently, the ratio between {M_2_O_3_} and {Li_2_O} was adjusted, as well as the internal composition of M. Four possible formulations were identified.

The 3-valence cation was set to M = Li_0.28_ Na_0.031_ B_0.016_ Al_0.063_ Si_0.578_ Zr_0.016_ P_0.016_. The specific composition was then experimentally adjusted to obtain the M formulation with a {M_2_O_3_}:{Li_2_O} ratio of 14:2. The average cation was M_28/32_Li_4/32_ and the average cation valence was 2.75.The 3-valence cation was also set to M = Li_0.28_ Na_0.031_ B_0.016_ Al_0.063_ Si_0.578_ Zr_0.016_ P_0.016_, and the {M_2_O_3_}:{Li_2_O} ratio was set to 15:1 with an average cation valency of 2.875. The average cation was M_30/32_Li_2/32_, with a change in the structural unit proportion. The M formulation was unchanged with respect to the C1 formulation, which increased the average valency, that is, a reduced amount of precipitated lithium-rich phases.The 3-valence cations were set to M = Li_0.312_ Na_0_ B_0.016_ Al_0.063_ Si_0.578_ Zr_0.016_ P_0.016_, and the {M_2_O_3_}:{Li_2_O} ratio was set to 15:1 with an average cation valency of 2.875. The average cation concentration was M_30/32_Li_2/32_. The Na content was reduced to 0, whereas that of Li increased with respect to the C2 composition (by adjusting the Li/Na ratio).The 3-valence cations were set to M = Li_0.25_ Na_0.062_ B_0.016_ Al_0.063_ Si_0.578_ Zr_0.016_ P_0.016_, and the {M_2_O_3_}:{Li_2_O} ratio was set to 14:2 with an average cation valency of 2.75. The average cation concentration was M_28/32_Li_4/32_. The Na content increased with respect to the C2 composition, whereas Li content reduced (by adjusting the Li/Na ratio).

### 3.2. Preparation of Glass-Ceramics

The composition design of Li_2_O–Al_2_O_3_–SiO_2_ glass was as shown in Table 1. Each composition corresponding to the raw materials were SiO_2_, H_3_BO_3_, and NH_4_H_2_PO_4_ (purity: 98%; XiLONG SCIENTIFIC, Guangzhou, China); Li_2_CO_3_, Na_2_CO_3_, and Al_2_O_3_ (purity: 99%; Macklin, Shanghai, China); ZrO_2_ (purity: 99%; Alladdin, Shanghai, China). The above raw materials were weighed and mixed to form a glass batch with a weight of 200 g. The batch was thoroughly mixed for 20 min using an automatic blender, transferred to a fused silica crucible, and preheated for 30 min in a high-temperature furnace that was heated to 1100 °C. The glasses were then slowly heated to 1520 °C at a rate of 10 °C/min to completely melt the compound. To ensure uniformity and clarity, the glasses were maintained at 1520 °C for 2 h. The molten glasses were then poured into a mold for rapid cooling to form a block. The mold was preheated to 200 °C, and the block was annealed in a muffle furnace at 450 °C for 4 h and then cooled to 25–30 °C to relieve internal stress.

### 3.3. Structure and Performance Test

The non-isothermal kinetic parameters of the crystals were determined using differential scanning calorimetry (DSC). The DSC analysis was performed using approximately 30 mg of fine glass powder (particle diameter of 1–10 µm), which was analyzed using a NETZSCH STA 449 F5 thermal analyzer (Bavaria, Germany). The powder was placed in a fused silica crucible and tested by increasing the temperature at a 10 °C/min rate. The test temperature range was 30 to 1000 °C.

An XRD analysis of the crystallized glasses was performed by grinding the glass-ceramics into a 200-mesh powder and using an X-ray diffractometer (SmartLab (3 kW), Rigaku, Tokyo, Japan) to analyze the physical phases of the powdered glass-ceramic samples in the range of 10–80°. The X-ray diffractometer was equipped with graphite-monochromator-filtered Cu Kα radiation (λ = 0.154 nm) at 40 kV and 40 mA. The divergence, receiving, and scattering slits were set at 1°, 0.15 mm, and 1°, respectively. The scanning speed was 10°/min. The results were used to calculate the degree of crystallinity, and the relative content of the crystalline phases was determined using MDI Jade 6.5 software (Materials Data (MDI), Livermore, CA, USA). 

The lithium aluminosilicate glass-ceramic samples were etched with 5% hydrofluoric acid for 50 s, rinsed with deionized water, dried, sputted with gold, and the images were recorded via scanning electron microscopy (SEM; TEScould MIRA3, Brno, Czech Republic) to investigate the surface morphology of the samples. During the analysis, the accelerating voltage was set at 20 kV.

After the heat treatment process, the glass-ceramics were ground into a 200-mesh powder using an XPM-Φ120 × 3 3-head grinder. Approximately 1 mg of glass-ceramic powder was added to KBr powder for mixing and grinding and was poured into the mold for pressing. An American Thermo, Nicolet iS50 Fourier transform infrared (FT-IR) spectrometer was used to analyze the phase of pressed tablet samples. The wavelength range was 400–1000 nm, and the test was performed at 24 °C.

The bending strength test was performed using the 3-point bending method. The glass-ceramics were cut into strips with dimensions of 2 × 5 × 40 mm^3^ and polished. Bending strength testing was performed using a Yuhan Model YC-128A (Shanghai, China) universal mechanical testing machine, and pressure was applied to the sample until a fracture occurred. The experimental span was set to 25 mm, the loading speed was 9.8 ± 0.1 Ns-1, and the sample size of the glass-ceramic was used. Each sample was tested 20 times to ensure the accuracy of the experiment and to avoid mechanical errors, and to determine the standard deviation of the data. The crystallized 10 × 10 × 1 mm^3^ glass piece was roughly polished, and its hardness was measured at 5 points using a Taiming HXD-1000TMC/LCD (Shanghai, China) microhardness tester, with an experimental loading time of 10 s and a loading pressure of 1.96 N. The 5 Vickers hardness values were then averaged. The PerkinElmer Lambda 650 (Waltham, MA, USA) ultraviolet-visible spectrophotometer was used to test the visible light transmission rate of glass-ceramics. The crystallized 10 × 10 × 1 mm^3^ glass pieces were finely polished and then tested in the wavelength range of 380–800 nm at room temperature.

## 4. Results and Discussion

### 4.1. DSC

Figure 2 shows the DSC diagram of the base glasses for the lithium aluminosilicate system. The characteristic temperature points of the DSC curves are listed in Table 2, where *T*_g_ is the glass transition temperature and *T_p_*_1_ and *T_p_*_2_ are the first and second crystallization peak temperatures, respectively. As shown in Figure 2, the glass transition temperature of four samples is in the 500–540 °C range; the glass starts to be substantially adjusted, and nucleation occurs in this temperature range. The crystallization peak temperatures of the four samples are also analyzed; *T_p_*_1_ and *T_p_*_2_ are found to be in the range of 660–700 °C and 720–770 °C, respectively. In this temperature range, fine grains grow in the glass. Near 860 °C, the curves of C2 and C4 base glasses show tiny exothermic peaks, but no new crystal phase can be found after holding this temperature for 4 h. In the range of 920–950 °C, four DSC curves all show endothermic peaks, and at this time, the crystal begins to melt in this temperature range.

As can be seen in Figure 2 and Table 2, the lower *T_g_* values for the C1 and C4 glass samples are due to the higher content of alkali metal oxides (total of Na_2_O and Li_2_O) in these two samples, which can significantly modify the glass network. The alkali metal ions provide "free oxygen" in the system, which increases the O/Si ratio in the glass network structure and leads to the declustering of the siloxane anion clusters into simple structural units in the silicate network structure. Therefore, the increase in alkali metal oxides reduces the activation energy of the glass and the viscosity of the glass melt compared with the C2 and C3 samples, making it easier for the parent glass to precipitate the crystalline phase. The DSC curve of the C3 sample shows that the temperature difference between the two crystallization peaks is significantly smaller than that of the other samples, while the C4 sample has the largest temperature difference between the crystallization peaks. This is due to the fact that the C3 sample does not contain Na_2_O, while the C4 sample has more Na_2_O added to replace some of the Li_2_O. Compared to Li^+^, the polarization ability of Na^+^ is weaker and has less effect on breaking the Si–O bond, which makes T_p2_ move to higher temperatures. 

Preliminary experiments show that when nucleation and crystallization heat treatments are performed at a lower temperature, the interior of the glass-ceramic tends to precipitate fine nanosized and uniformly dispersed crystal particles [35]. Therefore, by analyzing the DSC curves, it can be inferred that the heat treatment system of all the glass-ceramics will adopt a two-step method: the parent glass is pretreated at 570 °C for one hour, and the heating rate is 10 °C/min; the nucleated glass is crystallized at 760 °C for two hours at the same heating rate. After the melting time, the furnace was slowly cooled to room temperature. The mother glass and the glass-ceramic pictures are shown in Figure 2. According to other studies [1,2,5,6,7,35], the main crystalline phase of most lithium aluminosilicate glass-ceramics is the β-quartz solid solution phase, and the lithium aluminosilicate glasses in this study have lower melting, nucleation, and crystallization temperatures compared with the results of others.

### 4.2. Structural Analyses 

#### 4.2.1. Infrared (IR) Spectroscopy Analysis

The fine structure of the glass-ceramics was investigated using FT-IR spectroscopy. The characteristic peaks of all four lithium aluminosilicate glass-ceramics were concentrated in the wavenumber range of 400–1000 cm^−1^ (Figure 3). According to the relevant literature [36], the vibrational spectra of inorganic glasses predominantly depend on the glass network formers, and the effect of glass network modifiers on the vibrational spectra is relatively minor.

The energy band at approximately 430 cm^−1^ was due to either the bending vibration of the O–Si–O bond [37] or the symmetric stretching vibration of the [Li–O_4_] tetrahedron. The energy band at 461 cm^−1^ was attributed to the Si–O–Si bending vibration. In addition, all shelf silicate minerals exhibited moderate intensity bands between 550 and 850 cm^−1^, which is a natural consequence of their polymeric structure. In this range, Si–O–Al bending vibrations (560 cm^−1^) and Al–O–Si symmetric stretching vibrations (640 cm^−1^) are observed [35]. This indicates that Al atoms entered the [Si–O_4_] tetrahedra (partially replacing Si atoms). The Al–O bonds in the [Al–O_6_] octahedra (possibly with the [Si–O_4_] tetrahedral co-prismatic Al–O bonds) were likely transformed into the [Al–O_4_] tetrahedra. This phenomenon is due to the replacement of Si^4+^ ions by Al^3+^ ions in the lattice; thus, Li^+^ ions fill the vicinity of Al^3+^ ions for charge neutrality. The IR peaks at 765 and 793 cm^−1^ were attributable to the symmetric stretching vibrations of O–Si–O and Si–O–Si, respectively [8]. The peaks between 930 and 941 cm^−1^ are characteristic of the lithium disilicate structure [38].

In addition to the characteristic vibrations discussed above, a moderate-intensity band located at approximately 850 cm^−1^ was observed in the C1 sample, which was attributable to the characteristic vibrations of the nonbridging oxygen [Al–O_6_] octahedra [39]. This band was not observed for the other samples, indicating that some of the [Al–O_6_] octahedra were transformed into [Al–O_4_] tetrahedra. The remaining [Al–O_6_] octahedra were not connected to the lithium aluminosilicate glass network. As a result, the homogeneity of lithium aluminosilicate glasses is reduced [40]. All the samples exhibited a strong absorption band in the 720–780 cm^−1^ region, which indicates that this band corresponds to the characteristic vibration of the Al–O covalent bond in the [Al–O_4_] tetrahedra present within the perovskite feldspar [41]. The Si–O–Al bending vibration (560 cm^−1^), Al–O–Si symmetric stretching vibration (640 cm^−1^), O–Si–O symmetric stretching vibration (765 cm^−1^), and Si–O–Si (793 cm^−1^) symmetric stretching vibrations were significantly weaker in the C1 sample than in the other samples. This is probably due to the C1 formulation containing a lower amount of O^2−^ ions than the other samples, and therefore, it cannot form more oxygen bridges.

#### 4.2.2. XRD

The properties of glass-ceramics depend on the percentage of crystalline phases formed, the type, composition, shape, and size of the crystals, and the residual glassy phase composition [35]. As shown in Figure 4, all four samples exhibit Li_2_Si_2_O_5_ (JCPDS #40-0376) and LiAlSi_4_O_10_ (JCPDS #35-0463) crystalline phases, and the crystallinity and crystal proportion of the four lithium aluminosilicate glasses differ according to the MDI Jade 6.5 analysis.

The crystallinities of samples C1, C2, C3, and C4 were 32.3%, 85.6%, 74.9%, and 61.1%, respectively. The proportions of the two main crystalline phases of the four samples were as follows: C1:Li_2_Si_2_O_5_:LiAlSi_4_O_10_ = 92.3:7.5; C2:Li_2_Si_2_O_5_:LiAlSi_4_O_10_ = 80.1:19.6; C3:Li_2_Si_2_O_5_:LiAlSi_4_O_10_ = 84.2:14.5; and C4:Li_2_Si_2_O_5_:LiAlSi_4_O_10_ = 89.5:9.2. LiAlSi_4_O_10_ is a type of framework silicate mineral with a Vickers hardness of >930 Hv, whereas the Vickers hardness value of Li_2_Si_2_O_5_ is >1100 Hv. The mechanical properties of Li_2_Si_2_O_5_ and LiAlSi_4_O_10_ were considerably improved after Li_2_Si_2_O_5_ and LiAlSi_4_O_10_ were precipitated from lithium aluminosilicate glasses.

According to the model, the C1 and C4 formulations were {M_2_O_3_}:{Li_2_O} = 14:2, and the average cation valence of the base glasses was 2.75. The addition of Al_2_O_3_ was not sufficient to match the Li_2_O content, resulting in the formation of a certain number of LiAlSi_4_O_10_ crystals. This results in less than 10% of the precipitated crystals being LiAlSi_4_O_10_ crystals, while Li_2_Si_2_O_5_ crystals account for over 90% of the precipitated crystals. The glass-ceramics exhibited large grain sizes and low degrees of crystallinity. The average cation valence of samples C2 and C3 was 2.875, the proportion of LiAlSi_4_O_10_ in the crystals increased significantly, the crystallinity increased considerably, and the overall grain sizes are obviously smaller than that of C1 and C4 samples.

#### 4.2.3. Field Emission SEM

To determine the crystal size and morphology of the investigated glass-ceramics and lay a good foundation for further analysis of the relationship between the degree of crystallization and the resulting properties, field emission SEM (FESEM) imaging was conducted [35]. The crystal morphologies of the four lithium aluminosilicate glass-ceramics are shown in Figure 5.

The FESEM image of the original C1 component shows that its grain size is in the range of 200–400 nm. In particular, most grains are approximately 200 nm in size and are spherically shaped. Furthermore, few grains are rod-shaped; the rod-shaped grains are nonuniformly distributed around the spherical crystals, and there are large gaps between the grains. Usually, the Li_2_Si_2_O_5_ crystal is rod-shaped, but in our study, it is mainly spherical. According to the research content in our previous published article [42], the crystals precipitated from this series of lithium aluminosilicate glass show three-dimensional growth. According to the CLSM image of this paper, a large number of crystal nuclei are formed in the nucleation stage, and the intervals between them are very small, which means there is not enough room for the growth of grains in a certain direction, and finally forms a nearly spherical crystal. Samples C2, C3, and C4 exhibited spherical grains. The grain size of sample C4 was approximately 150 nm and the grains were sparsely arranged. The grain size of sample C3 was approximately 120 nm, with the grains being sparsely arranged at an interval of approximately 50 nm. Sample C2 had fine grains approximately 80–100 nm in size; these grains were densely arranged and exhibited the highest degree of crystallization. The grain sizes of the four types of glass-ceramics are in the order of C1 > C3 > C4 > C2.

### 4.3. Mechanical Properties and Light Transmittance

#### 4.3.1. Microhardness and Flexural Strength

Figure 6a shows the bending strength and microhardness of the four lithium aluminosilicate glass-ceramics. In order to clarify the advantages and disadvantages of the mechanical and optical properties of glass-ceramics, its performance is compared with that of a certain high aluminosilicate glass G. High aluminosilicate glass G is the mainstream cover glass for touch screen devices in the market today. The curves show that the hardness and bending strength from high to low of the samples can be arranged as C2, C3, C4, and C1. Compared with the C1 sample, the flexural strength of the C2 sample increases by 37.9% and the microhardness increases by 51.4%. From C2 to C1, the crystallinity of the samples decreased but the grain size increased. It can be seen that degree of crystallinity and crystal size affect the mechanical properties of glass-ceramics.

The microhardness of glass-ceramics is strongly related to the internal glass phases and crystal structure, both of which dictate the overall performance of glass-ceramics. On the one hand, the glass phases consist of the matrix phases that bond the microcrystalline phases together to form a dense structure. In contrast, the microcrystalline phases strengthen the structure and glass phases, and the two complement each other. The FESEM image shows that several glass-ceramic crystals in the base glasses are spherical or nearly spherical and do not form mechanical structures, such as a building block structure or needle interlocking structure. Therefore, the increase in the proportion of the microcrystalline phases is the most important factor affecting the microhardness of the glass-ceramics, which is significantly increased. The XRD and hardness results of this study indicate that the degree of crystallinity and hardness were the highest for sample C2, followed by those of samples C3, C4, and C1; however, for all samples, they were higher than those of high-aluminosilicate glasses.

As shown in Figure 6a, the situation changes significantly after the microcrystallization of the base glasses. Owing to the presence and content of the crystalline phases, the microstructure of the glass-ceramics is improved, and equivalent to adding a reinforcing or toughening phase to the original pure glass phases. Furthermore, the bending and fracture mechanism of the material becomes more complex, and the bending and fracture resistance of the investigated glasses are significantly higher (up to 159 MPa) than those of high-aluminosilicate glasses.

#### 4.3.2. Transmittance

The transmittance of the glass-ceramics was determined by calculating the ratio of the refractive index of the crystalline phases to the glass phases and crystal sizes. Kerker [43] reported that the loss due to light scattering is extremely low when the size of the microcrystals formed in base glasses is less than 15 nm and the refractive index difference *n* is less than 0.1. Hopper [44] reported that the spacing between the microcrystals in glass-ceramics is a value greater than the radius of the microcrystals less than six times the radius, that is, satisfying the boundary conditions of dense microcrystalline scattering theory.

The refractive index of the glass phases and LiAlSi_4_O_10_ was approximately 1.5, the refractive index of Li_2_Si_2_O_5_ was approximately 1.55, the refractive index ratio of LiAlSi_4_O_10_ to the glass phases was approximately 1, and the refractive index ratio of Li_2_Si_2_O_5_ to the glass phases was approximately 1.03. Thus, the refractive index difference was only 0.03%, and the grain size was a factor affecting the transmittance. Glass-ceramics C1~C4 and high aluminosilicate glass G with a thickness of 1mm were prepared, and their visible light transmittance was tested. From the FESEM images and Figure 6b, it can be seen that the grain size of sample C2 is the smallest, followed by those of samples C3, C4, and C1, whereas the transmission rate follows the opposite trend. The most favorable optical performance of sample C2 is similar to that of high-aluminosilicate glasses in the main wavelength bands and meets the requirements of touch screen covers, windshields, and other applications.

## 5. Conclusions

Using the cluster-plus-glue-atom model, the C1 base glass formulation was optimized to obtain C2–C4 formulations. The average valence of the C1 and C4 cations was 2.75. Conversely, the average valence of the C2 and C3 glass cations was 2.875 after optimization, resulting in glass-ceramics with higher crystallinity and superior properties.The main crystalline phases of the lithium aluminosilicate glass-ceramics prepared in this work were Li_2_Si_2_O_5_ and LiAlSi_4_O_10_. The mechanical properties of lithium aluminosilicate glasses were significantly enhanced by the presence of these two crystalline phases, with a 37.9% increase in bending strength and a 51.4% increase in microhardness. The refractive indices of the two crystalline phases are close to that of the base glasses, and the glass-ceramics were transparent when the grain size was less than 100 nm. Compared with the C1 sample, the C2, C3, and C4 samples had higher crystallinity after heat treatment. The crystal size was in the range of 100–200 nm, which resulted in a significant improvement in the optical and mechanical properties.When the C2 formulation was selected to melt the base glasses, it nucleated at 570 °C for one hour and crystallized at 660 °C for two hours. The resulting glass-ceramics exhibited a degree of crystallinity that reached 90%, as well as excellent properties, with a visible light transmission above 90%, a bending strength of 159 MPa, and a microhardness of 967 Hv.

## Figures and Tables

**Figure 1 nanomaterials-13-00530-f001:**
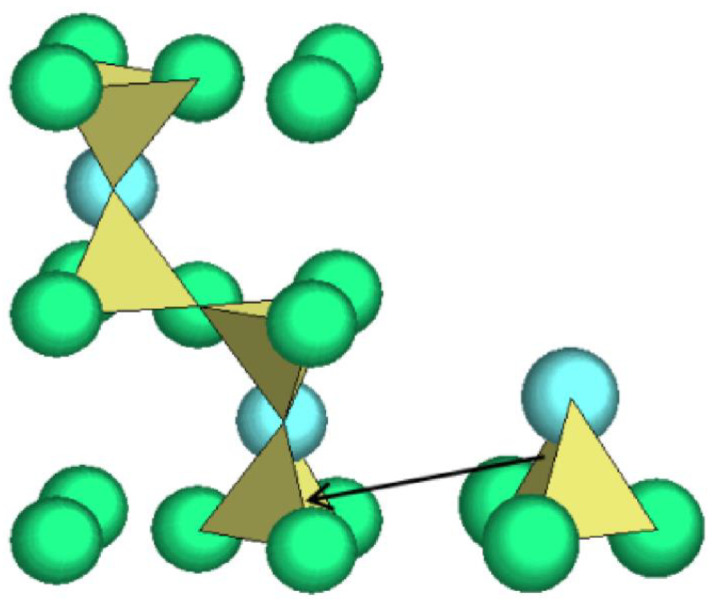
Crystal structure of beta-tridymite using tetrahedral cluster [Si–O_4_].

**Figure 2 nanomaterials-13-00530-f002:**
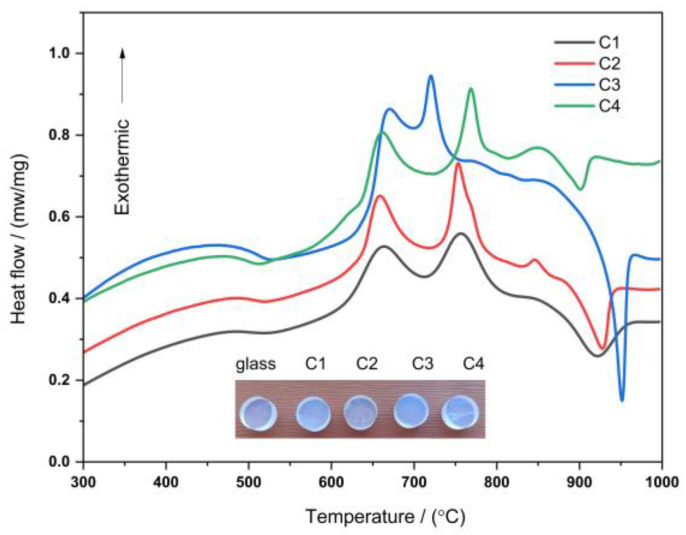
DSC curve of C1~C4 lithium aluminosilicate glasses and pictures of glass and glass-ceramics.

**Figure 3 nanomaterials-13-00530-f003:**
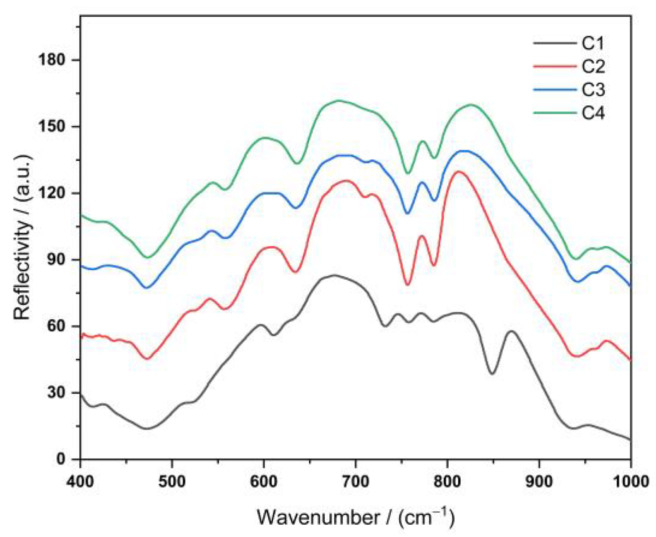
FT-IR spectrum of lithium aluminosilicate glass-ceramics with four different formulations.

**Figure 4 nanomaterials-13-00530-f004:**
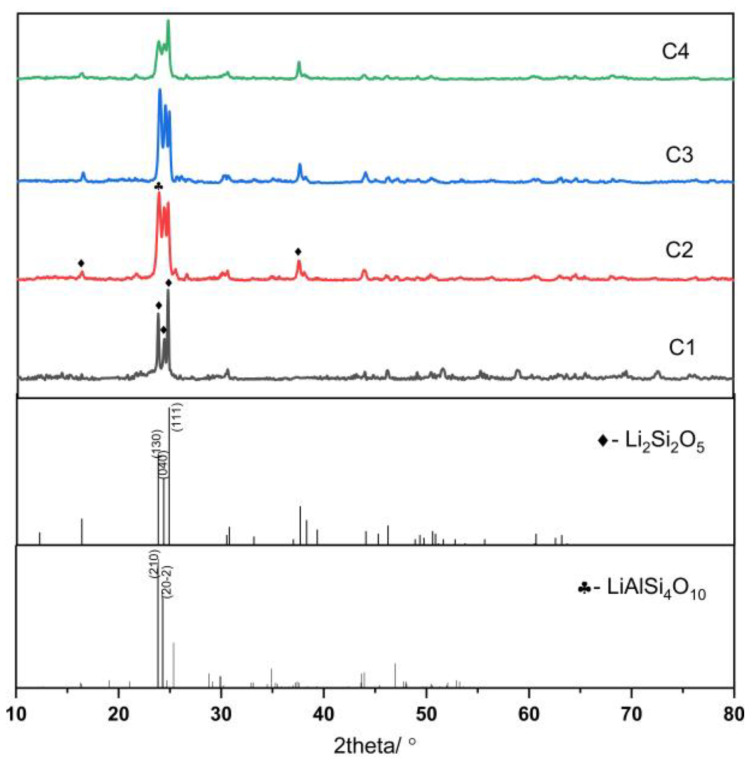
XRD pattern of lithium aluminosilicate glass-ceramics.

**Figure 5 nanomaterials-13-00530-f005:**
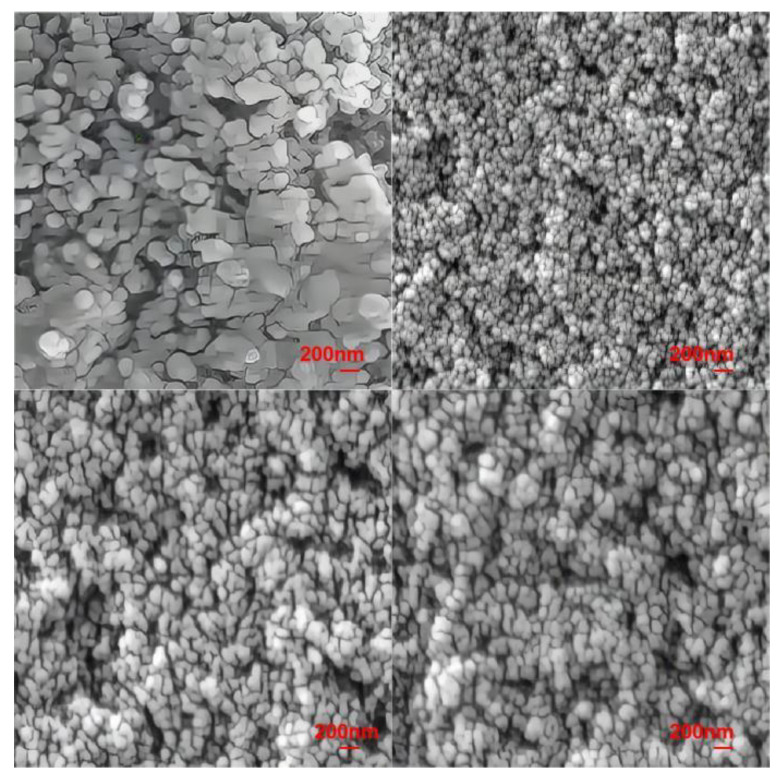
FESEM images of lithium aluminosilicate glass-ceramics.

**Figure 6 nanomaterials-13-00530-f006:**
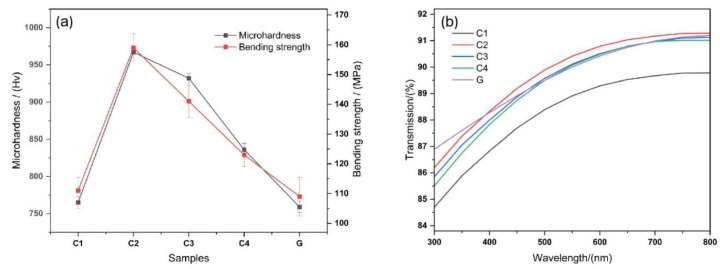
(**a**) Schematic of the flexural strength and microhardness of lithium aluminosilicate; (**b**) visible light transmission curve of lithium aluminosilicate glass-ceramics.

**Table 1 nanomaterials-13-00530-t001:** Composition design of Li_2_O–Al_2_O_3_–SiO_2_ system glass-ceramics (mol%).

Formulation (mol.%)	SiO_2_	Al_2_O_3_	Na_2_O	Li_2_O	B_2_O_3_	P_2_O_5_	ZrO_2_
C1	66.58	3.60	1.80	24.42	0.90	0.90	1.80
C2	69.64	3.76	1.88	20.95	0.94	0.94	1.88
C3	69.64	3.76	0	22.83	0.94	0.94	1.88
C4	69.64	3.76	3.76	19.07	0.94	0.94	1.88

**Table 2 nanomaterials-13-00530-t002:** DSC curves of the lithium aluminosilicate glasses (°C).

Recipes	C1	C2	C3	C4
*T_g_*	520	525	535	515
T_x_1	620	635	645	635
T_p_1	655	660	670	660
T_x_2	710	725	700	740
T_p_2	760	750	720	770

## Data Availability

Data openly available in a public repository.

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
