# Peer review of "Light-Transmitting Lithium Aluminosilicate Glass-Ceramics with Excellent Mechanical Properties Based on Cluster Model Design"

_nanomaterials, 2023, doi:10.3390/nano13030530_

Round 1

Reviewer 1 Report

Review attached

Author Response

Reviewer #1: Thanks to the professional opinions of the reviewers. According to your comments, I revised the article and answered the questions. The font color of the modified content is red in the article and this reply. Due to a large number of revisions in this paper, the original number of lines has changed, and the new revision position has been marked.

I found the manuscript: “Excellent mechanical properties and light-transmitting lithium

aluminosilicate glass–ceramics based on cluster model design” to be overall very well

informed, objective, comprehensive, accurate, and well written for the intended audience. I think that it is largely acceptable, but I have a number of comments for the authors’ consideration.

  1. The abstract was badly written. Practically, verses 9 to 16 can be moved to the

Introduction. The abstract should contain brief information about the results and

research methods.

Thank you for your comments. We have deleted unnecessary content and revised the abstract as follows.

“In this study, for the first time, a cluster-plus-glue-atom model was used to optimize the composition of lithium aluminosilicate glass–ceramics. Basic glass of glass–ceramics were considered to be a 16-unit combination of three-valence {M2O3} and one-valence {Li2O} units. By adjusting the ratio of {M2O3} and {Li2O} , the composition of basic glass could be optimized. After optimization, the average cations valence of the base glass were increased to 2.875. After heat treatment of the optimized base glass, it is found that the crystal size, proportion and crystallinity changed obviously compared with that before optimization. The main crystalline phases of all the lithium aluminosilicate glass–ceramics prepared in this work were Li2Si2O5 and LiAlSi4O10. All optimized glass-ceramics had an obvious improvement in the crystallinity, with one of the largest having a crystallinity of over 90%. Furthermore, its bending strength was 159 MPa, the microhardness was 967 Hv, the visible-light transmission rate exceeded 90%. Compared with the widely used touch panel cover glass, the optical properties were close and the mechanical properties were greatly improved.”

  1. The references were written very carelessly. Guidelines for authors were not followed.

All the references have been revised according to the unified format standard.

The revision of references has been carried out in lines 441-526 of the revised manuscript.

  1. In the section preparation of glass-ceramics it says that the glasses were heated in a quartz crucible (line 178). Are the authors sure that the crucibles were quartz?

In order to express accurately, we have changed quartz crucible into fused silica crucible (line 182、183 and 195). We put the pictures and their technical function index of quartz crucible used below.

  1. In what temperature range were DSC measurements performed? Authors should add this information in the paragraph describing the DSC method.

Thank you for your comments. We added the temperature range of DSC test in line 195-196 of this article. Add Sentence is “The test temperature range is 30 to 1000℃.”

  1. Why do the authors not comment on the thermal effects above 800 degrees Celsius

recorded on the DSC curves?

Thank you for reminding me thatthermal effects above 800℃ has been added in line of 242~245. “Near 860℃, the curves of C2 and C4 base glasses show a tiny exothermic peaks, but no new crystal phase can be found after holding at this temperature for 4 hours. In the range of 920~950℃, four DSC curves all show endothermic peaks, and at this time, the crystal begin to melt in this temperature range.”

The manuscript is worth publishing in Nanomaterials after addressing the above comments.

Reviewer 2 Report

I   found this article interesting. However, to fit the journal's requirements, the manuscript has to be significantly revised, in particular, the introductory chapter.  

  1. Line 31. The introduction chapter starts with the following statement:

"Glass–ceramics are inorganic materials that combine the properties of glasses and ceramics." However, it is not a definition  of glass ceramics.

More correct wording is following.

"Glass ceramics is a polycrystalline solid obtained due to controlled crystallization of the glass."

In a present form manuscript touches very narrow topic.  To show the importance of lithium silica ceramics in comparison with other options, for instance, barium silica ceramics. I recommend that authors look through a few of their previous publications.

1.     Beall GH (1992) Design and properties of glass-ceramics. Annu Rev Matter Sci 22: 91‒119

2.     W.Holand, G.H.Beall, Glass ceramics technology, Second edition, Wiley, 2012

3.     Deubener J (2004) Configurational entropy and crystal nucleation of silicate glasses. Phys Chem Glass 45:61

4.     Bliss M, Reeder PL, Weber MJ, Craig RA, Sunberg DS (1994) Relationship between microstructure and efficiency of scintillation glasses, PNL-SA-23185, April 1994

5.     Y. Tratsiak, A. Fedorov, G. Dosovitsky, O. Akimova, E. Gordienko, M. Korjik, V. Mechinsky, E. Trusova. Scintillation efficiency of binary Li2O-2SiO2 glass doped with Ce3+ and Tb3+ ions. Journal of Alloys and Compounds Vol. 735 (2018), pp. 2219–2224. DOI https://doi.org/10.1016/j.jallcom.2017.11.386

6.     Tratsiak Y., Korjik M., Fedorov A., Dosovitsky G., Akimova O., Gordienko E., Fasoli M., Mechinsky V., Vedda A., Moretti F., Trusova E. Luminescent properties of binary MO-2SiO2 (M = Ca2+, Sr2+, Ba2+) glasses doped with Ce3+, Tb3+ and Dy3+ // Journal of Alloys and Compounds. – 2018. – Vol. 765. – P. 207

7.     Lecoq, P.; Gektin, A.; Korzhik, M. Inorganic Scintillators for Detector Systems; Particle Acceleration and Detection; Springer International Publishing: Cham, 2017; ISBN 978-3-319-45521-1 *Chapter 9)

Chapter 3.2

This chapter describes  the preparation of the mother glass. It is important to describe the regime of controlled crystallization. It would be important to show an image of the mother glass and the glass ceramics. The surface crystallization should be commented on as well, if it appears. The fraction of crystallites in the mother glass should be estimated.

Fig.2. This figure has to be compared with the results of others. A distinction and modification of the DSC curve on the sample must be discussed in greater depth.

Fig.5. The SEM images look a bit unusual. They have to include amorphous phase and crystallites. Glass can contain bubbles  but quite rarely pores, as seen in fig.5 in the left upper corner. The spherical habitus of the lithium disilicate particles has to be commented, it is different from what is known from the literature.

Fig.6 (b).  The thickness  of the samples should be indicated. It would be more useful if the spectral dependence of the absorption coefficient will be presented. The comparison of absorption and transmission spectra gives an immediate impression of the contribution of the scattering  or absorption (color centers) of the defects.

Manuscript required major revision.

Author Response

Reviewer #2: Thanks to the professional opinions of the reviewers. According to your comments, I revised the article and answered the questions. The font color of the modified content is blue in the article and this reply. Due to a large number of revisions in this paper, the original number of lines has changed, and the new revision position has been marked.

I found this article interesting. However, to fit the journal's requirements, the manuscript has to be significantly revised, in particular, the introductory chapter.  

1、Line 31. The introduction chapter starts with the following statement:

"Glass–ceramics are inorganic materials that combine the properties of glasses and ceramics." However, it is not a definition of glass ceramics.

More correct wording is following.

"Glass ceramics is a polycrystalline solid obtained due to controlled crystallization of the glass."

Thank you for your comments. We have modified and replaced the sentence in line 26-29 of this article.The revised sentence is as follows:

“Glass ceramics is a polycrystalline solid obtained due to controlled crystallization of the glass. They are obtained by uniformly precipitating numerous tiny crystals in a glass matrix to form a multiphase composite comprising dense crystalline phase and glass phases.”

2、In a present form manuscript touches very narrow topic.  To show the importance of lithium silica ceramics in comparison with other options, for instance, barium silica ceramics. I recommend that authors look through a few of their previous publications.

  1. Beall GH (1992) Design and properties of glass-ceramics. Annu Rev Matter Sci 22: 91‒119
  2. Holand, G.H.Beall, Glass ceramics technology, Second edition, Wiley, 2012
  3. Deubener J (2004) Configurational entropy and crystal nucleation of silicate glasses. Phys Chem Glass 45:61
  4. Bliss M, Reeder PL, Weber MJ, Craig RA, Sunberg DS (1994) Relationship between microstructure and efficiency of scintillation glasses, PNL-SA-23185, April 1994
  5. Tratsiak, A. Fedorov, G. Dosovitsky, O. Akimova, E. Gordienko, M. Korjik, V. Mechinsky, E. Trusova. Scintillation efficiency of binary Li2O-2SiO2 glass doped with Ce3+ and Tb3+ ions. Journal of Alloys and Compounds Vol. 735 (2018), pp. 2219–2224. DOI https://doi.org/10.1016/j.jallcom.2017.11.386
  6. Tratsiak Y., Korjik M., Fedorov A., Dosovitsky G., Akimova O., Gordienko E., Fasoli M., Mechinsky V., Vedda A., Moretti F., Trusova E. Luminescent properties of binary MO-2SiO2 (M = Ca2+, Sr2+, Ba2+) glasses doped with Ce3+, Tb3+ and Dy3+ // Journal of Alloys and Compounds. – 2018. – Vol. 765. – P. 207
  7. Lecoq, P.; Gektin, A.; Korzhik, M. Inorganic Scintillators for Detector Systems; Particle Acceleration and Detection; Springer International Publishing: Cham, 2017; ISBN 978-3-319-45521-1 *Chapter 9)

Thank you for your suggestion, I have read the relevant papers and books. Since our team will subsequently publish many research results on the application of the cluster-plus-glue-atom model for design and optimization of the composition of different glass systems. Therefore, in this paper, only a brief description of the use of the model in other systems will be given in line 135-140 of this article. The added de content is as follows. 

“For other systems of glass and glass-ceramics, the model is used in the same way. Such as CaO-2SiO2, SrO-2SiO2, BaO-2SiO2 [30-33], CaO-MgO-SiO2 [34] and so on. Taking CaO-MgO-SiO2 glass-ceramics as an example, it can be considered to be a 16-unit combination of three-valence {M2O3} and two-valence {Mg2O2} units for design and optimization.”

3、Chapter 3.2

This chapter describes the preparation of the mother glass. It is important to describe the regime of controlled crystallization. It would be important to show an image of the mother glass and the glass ceramics. The surface crystallization should be commented on as well, if it appears. The fraction of crystallites in the mother glass should be estimated.

We have added the mother glass and the glass-ceramics pictures in Figure 2. The nucleation and crystallization regime of the mother glass have been given in chapter 4.1 (line of 265-268).

The crystallization mechanism that form from this series of lithium aluminosilicate glass are three-dimensional growth, according to the research presented in our earlier paper. The paper is “42. Li, M,; Xiong, C.; Ma, Y.; Jiang H. Study on Crystallization Process of Li2O–Al2O3–SiO2 Glass-Ceramics Based on In Situ Analysis. Materials. 2022, 15, 8006-8017.”

4、Fig.2. This figure has to be compared with the results of others. A distinction and modification of the DSC curve on the sample must be discussed in greater depth.

In the chapter 4.1 of the article(line of 246-260, 269-272) , we have added the in-depth analysis of the DSC curve of lithium aluminosilicate glass studied and the comparison with other people's research results. The added contents are as follows.

“As can be seen in Fig. 2 and Table 2, the lower Tg values for the C1 and C4 glass samples are due to the higher content of alkali metal oxides (total of Na2O and Li2O) in these two samples, which can significantly modify the glass network. The alkali metal ions provide "free oxygen" in the system, which increases the O/Si ratio in the glass network structure and leads to the declustering of the siloxane anion clusters into simple structural units in the silicate network structure. Therefore, the increase in alkali metal oxides reduces the activation energy of the glass and the viscosity of the glass melt compared to the C2 and C3 samples, making it easier for the parent glass to precipitate the crystalline phase [36]. The DSC curve of the C3 sample shows that the temperature difference between the two crystallization peaks is significantly smaller than that of the other samples; while the C4 sample has the largest temperature difference between the crystallization peaks. This is due to the fact that the C3 sample does not contain Na2O, while the C4 sample has more Na2O added to replace some of the Li2O. Compared to Li+, the polarization ability of Na+ is weaker and has less effect on breaking the Si-O bond, which makes Tp2 move to higher temperatures. “

“According to the other people's research analysis [1,2,5-7,35], the main crystalline phase of most lithium aluminosilicate glass–ceramics is β–quartz solid solution phase, and the lithium aluminosilicate glasses studied by us have lower melting, nucleation and crystallization temperatures compared with the results of others.

  • 5. The SEM images look a bit unusual. They have to include amorphous phase and crystallites. Glass can contain bubbles but quite rarely pores, as seen in fig.5 in the left upper corner. The spherical habitus of the lithium disilicate particles has to be commented, it is different from what is known from the literature.

Fig. 5 FESEM pictures has been changed. The previous SEM pictures showed that the glass matrix disappeared due to the long corrosion time of the glass-ceramics samples.

The explanation that the Li2Si2O5 crystal appears spherical has been added in line of 345-351 of this article. Specific content as follows.

“Usually, Li2Si2O5 crystal is rod-shaped, but in our study, it is mainly spherical. According to the research content in our previous published article [42], the crystals precipitated from this series of lithium aluminosilicate glass are three-dimensional growth. According to the CLSM image of this paper, a large number of crystal nuclei are formed in the nucleation stage, and the intervals between them are very small, which makes there not enough room for the growth of grains in a certain direction, and finally forms a nearly spherical crystal.”

6、Fig.6 (b).  The thickness of the samples should be indicated. It would be more useful if the spectral dependence of the absorption coefficient will be presented. The comparison of absorption and transmission spectra gives an immediate impression of the contribution of the scattering  or absorption (color centers) of the defects.

In the 228-231 line of the article, we have the size requirements for the transmittance test sample. “Using the PerkinElmer Lambda 650 (US) ultraviolet–visible spectrophotometer to test the visible light transmission rate of glass–ceramics. The crystallized 10 × 10 × 1 mm3 glass pieces were finely polished and then tested in the wavelength range of 380–800 nm at room temperature.”

We also added related sentences in line of 404-405 of this article. The added content is “Glass-ceramics C1~C4 and high aluminosilicate glass G with a thickness of 1mm were prepared, and their visible light transmittance was tested.”

   I'm very sorry, our university has been on holiday at present, and no one can take the test in the university. The test of absorption spectra and the comparison between absorption and transmission spectra can only be carried out after two months.

Manuscript required major revision.

Round 2

Reviewer 1 Report

The manuscript is suitable for publication in Nanomaterials.

Author Response

Thank the reviewers for agreeing to accept my manuscript.

Reviewer 2 Report

Now manuscript looks completed.

Line 345,  correct 

  1. Li2Si2O5

  1.  

Author Response

  • Line 345, correct: Li2Si2O5.

    The subscript of Li2Si2O5 has been modified as follows.

Usually, Li2Si2O5 crystal is rod-shaped, but in our study, it is mainly spherical. According to the research content in our previous published article [42], the crystals precipitated from this series of lithium aluminosilicate glass are three-dimensional growth. According to the CLSM image of this paper, a large number of crystal nuclei are formed in the nucleation stage, and the intervals between them are very small, which makes there not enough room for the growth of grains in a certain direction, and finally forms a nearly spherical crystal.
